# Printability and Tensile Performance of 3D Printed Polyethylene Terephthalate Glycol Using Fused Deposition Modelling

**DOI:** 10.3390/polym11071220

**Published:** 2019-07-22

**Authors:** Sofiane Guessasma, Sofiane Belhabib, Hedi Nouri

**Affiliations:** 1INRA, UR1268 Biopolymères Interactions Assemblages, F-44300 Nantes, France; 2Laboratoire GEPEA, UMR CNRS 6144, Université - IUT de Nantes, avenue du Professeur Jean Rouxel, 44475 Carquefou Cédex, France; 3IMT Lille-Douai, 941 rue Charles Bourseul, CS 10838, 59508 Douai, France; 4Laboratoires des Systèmes Electromécaniques (LASEM-ENIS), Université de Sfax, Route Soukra Km3, BPW3038, Sfax, Tunisia

**Keywords:** fused deposition modelling, polyethylene terephthalate glycol, tensile properties, X-ray micro-tomography, finite element computation

## Abstract

Polyethylene terephthalate glycol (PETG) is a thermoplastic formed by polyethylene terephthalate (PET) and ethylene glycol and known for his high impact resistance and ductility. The printability of PETG for fused deposition modelling (FDM) is studied by monitoring the filament temperature using an infra-red camera. The microstructural arrangement of 3D printed PETG is analysed by means of X-ray micro-tomography and tensile performance is investigated in a wide range of printing temperatures from 210 °C to 255 °C. A finite element model is implemented based on 3D microstructure of the printed material to reveal the deformation mechanisms and the role of the microstructural defects on the mechanical performance. The results show that PETG can be printed within a limited range of printing temperatures. The results suggest a significant loss of the mechanical performance due to the FDM processing and particularly a substantial reduction of the elongation at break is observed. The loss of this property is explained by the inhomogeneous deformation of the PETG filament. X-ray micro-tomography results reveal a limited amount of process-induced porosity, which only extends through the sample thickness. The FE predictions point out the combination of local shearing and inhomogeneous stretching that are correlated to the filament arrangement within the plane of construction.

## 1. Introduction

In fused deposition modelling (FDM), one of the versatile processes of additive manufacturing [1,2], the laying down of the polymeric material [3] occurs in a layer-by-layer basis according to complex toolpath trajectory [4].

The unidirectional laying down process results in two types of discontinuities; one is related to the raster within the plane of construction, and the other through the building direction. The combination of both discontinuities leads to a lack of structure cohesion within the 3D print and the genesis of 3D defects [5], which are essentially represented by a porosity network [6]. The effect of these defects can be significant on the mechanical performance that costs up one third of the performance of the feedstock material [6]. Several studies tackled this problem by adjusting key process parameters of FDM such as the part orientation, layer thickness [7], printing speed, and printing temperature [8,9,10,11]. This last parameter is known to enhance the cohesive structure of the 3D printed material by lowering the effect of necking and reducing the weight of the porosity effect on the mechanical performance [12,13]. Examples of studies that looked into the matter can be found in the literature for materials such as ethylene vinyl acetate [9], polylactic acid (PLA) [14], and acrylonitrile butadiene styrene (ABS) [15].

In this work, the influence of the extrusion temperature on the printability of polyethylene terephthalate glycol (PETG) is undertaken. Despite the availability of this copolyester in the market, few results are reported on the influence of the printing temperature on the thermal and mechanical behaviour of 3D printed PETG. This thermoplastic polymer of the formula (C_10_H_8_O_4_)_n_ can be formulated by the copolymerisation of PET ethylene glycol. PETG is thus a copolymer known for its high impact and chemical resistance, biocompatibility, transparency, and recyclability [16]. It can be used in different indoor applications (medical and food applications, electronics, …) with an acceptable flammability rating [17]. It has a low resistance to ultraviolet (UV) light and less performant against scratching and frictional contact. Szykiedans et al. [18] studied the mechanical performance of PETG and PETG reinforced with glass fibres printed using FDM technology. The authors highlighted the presence of air gaps within the prints and a large sensitivity of the mechanical performance to the printing orientation. Nieto et al. [19] considered pellet-based additive manufacturing route to print different polymers including the PETG. The authors observed a limited wrapping deformation and low shrinkage ratio of PETG. They explained such properties by the absence of crystallisation during the extrusion process. However, they pointed out difficulties to achieve a constant melt flow with PETG resulting in a particular texture for complex 3D printed parts. Hamidi et al. [20] reported printability results for the PETG reinforced with carbon nanotubes (CNT) using FDM technology. They observed a high distortion of the prints using this feedstock material. In addition, PETG + CNT proved to achieve a good bonding with steel fibres within the print and resulted in a low void content.

In this study, the printing temperatures for PETG are explored in a wide range typically from 220 °C to 255 °C. The effect of this process parameter on the thermal and mechanical behaviours is measured. In addition, finite element computation is considered to predict the influence of the defects induced by FDM on the mechanical performance taking into account the microstructural information issued from X-ray micro-tomography imaging.

## 2. Materials and Methods

The PETG filament is a polyester-based material provided by the FDBI company (Asnières sur seine, France) under the tradename 3DFilTech. The filament has a diameter of 1.75 ± 0.05 mm. The supplier recommends the use of a heated bed with typical temperatures between 40 °C and 65 °C. The supplier does not provide more technical data about the filament. Differential scanning calorimetry (DSC) experiments are performed to study the thermal behaviour of the as-received filament (Figure 1a). Two thermal scans are considered within a temperature range of (−40 °C to 300 °C) with a rate of 10 °C/min using DSC823e equipment from Mettler Toledo SAS (Viroflay, France). In addition, tensile performance on single filaments is performed using a universal testing machine (Zwick Roell, Ulm, Germany). The tensile experimental are conducted under a fixed displacement rate of 5 mm/min up to the rupture of the filament (Figure 1b).

The 3D printing process is undertaken using a commercial FDM machine (Replicator 2 from MakerBot, New York, NY, USA). All printing conditions are provided in Table 1. Among the selected conditions, room temperature is used for the bed. This choice is in line with the objective to print PETG using the lowest energy consumption. The printing temperature is varied between 210° up to the temperature limit for the printing machine (255 °C). The main form printed is a dog-bone specimen with typical dimensions shown in Figure 2a. The CAD (computer-aided design) model of this form is created according to the norm ISO 527-1/2. In addition, other types of specimens are printed including the one used for thermal analysis of the laying down process (Figure 2b). This geometry is used to allow studying the thermal cycling without being constrained by the movement of the printing nozzle. All printing stages including the preparation and processing of the CAD models are performed using the MakerBot© (New York, NY, USA).

The printing duration varies depending on the type of geometry between 7 min and 29 min. The density of dog-bone specimens is measured to derive the porosity content using the formula
(1)$$f\left(\%\right)=1-\left(\rho_{p}/\rho_{s}\right)$$
where f is the porosity content, $\rho_{s}$ and $\rho_{p}$ are the densities of the as-received and printed PETG.

The thermal cycling during the laying down process is measured using an infra-red camera (Flir A35 series from Flir company, Wilsonville, OR, USA). A full frame acquisition is considered (320 × 256 pixels) with a frame rate of 60 fps (frame per second).

X-ray micro-tomography imaging is performed on the dog-bone specimens using UltraTom X-ray micro-CT from Rx-Solutions (Chavanod, France) under a fixed voxel size of 14.31 µm and a detector resolution of 1920 × 1536 pixels. Up to 1440 radiographic images are needed to build the tomogram of the printed PETG using X-Act software from Rx-Solutions. The processed images represent a volume of 17.5 × 11.5 × 5.1 mm^3^, which is based on a resolution of 1224 × 804 × 354 voxels. The raw images are processed using ImageJ software from NIH to isolate the features of interest and measure the amount and distribution of the defects within the printed PETG.

In order to measure the possible loss in mechanical performance, tensile tensing is performed on dog-bone specimens using the same testing equipment and the same testing conditions introduced earlier. Notched specimens are also considered to measure the fracture toughness of PETG prints. The notch lengths represent from 8% to 12% of the sample width. From the engineering stress–strain curves, Young’s modulus, tensile strength, elongation at break, and fracture toughness are correlated to the printing temperature. All deformation sequences are recorded using optical camera (Phantom V7.3 from Photonline, Marly Le Roi, 78-France). The typical region of interest (ROI) varies from (144 × 200 pixels) up to (800 × 600 pixels) using an acquisition rate of 30 to 3000 fps (frame per second).

The fractured specimens are analysed using scanning electron microscope (JEOL JSM 7600F microscope, Tokyo, Japan) at different pixel sizes, typically from 0.5 µm (×190) to 3.1 µm (×30).

## 3. Modelling Technique

Finite element computation based on 3D images is implemented to predict the stress and strain distributions within the printed PETG and to test the fitness of the elasticity model by comparing the predicted and experimental Young’s modulus. The geometry is converted into a finite element mesh where each voxel is represented by a cuboid structural element. Each element is defined by eight nodes and each node has three degrees of freedom corresponding to the displacement components (UX, UY, UZ) in the main directions X, Y, and Z. The mesh density is adapted according to the resolution of the 3D images. Because computation resources are not sufficient to run models with meshes as large as the original resolution, a given collection of voxels is firstly converted into a single voxel (binning process), then secondly, the new voxel is converted into the cuboid element. Thus, the largest meshes that are achieved from the original resolution of the 3D images (1224 × 804 × 354 voxels) comprise 734 × 482 × 212 elements. This means that the total number of degrees of freedom (dof) that are solved is 163 million dof. With this model, the element size is 23.85 µm, and corresponds to 0.06 times the filament diameter. In order to check how the resolution lowering affects the quality of the FE predictions, smaller resolutions are considered down to 122 × 80 × 36 voxels, where the smallest resolution corresponds to an element size of 143 µm. The material model implemented is a linear elasticity model defined by Young’s modulus (791 MPa) and Poisson’s coefficient (0.4) of the as-received PETG filament. The simulation of the tensile test is performed by implementing the following boundary conditions: constrained end against displacement in all directions (UX = UY = UZ = 0) and displacement of the other end by a fixed amount U representing 1% of extension from the original length in the loading direction X (UX = +U, UY = UZ = 0).

The elasticity problem is solved iteratively using the preconditioned conjugate gradient (PCG). The predicted stress intensity field (σ_I_) and the effective Young’s modulus ($E_{f}$) are predicted. The simulation durations varies between 1 min and 219 min depending on the resolution. All computations are performed on a workstation equipped with 2 Xeon CPU operated at 3.0 Ghz and 1 Tbytes of RAM.

## 4. Results

### 4.1. Thermal Behaviour during Polyethylene Terephthalate Glycol (PETG) Laying Down

The DSC spectra corresponding to the heating and cooling stages of the as-received PETG highlights a temperature range between 42 °C and 111 °C corresponding to the glass transition (Figure 1a). According to the mid-point value from the second heating run, the temperature of glass transition of PETG is 77 °C. This value lies within the range (81 °C to 91 °C) reported from the data sheet. Oladapo et al. [21] report a glass temperature of 88 °C for PETG.

In terms of thermal behaviour during the laying down process of the PETG filament, Figure 3 shows a limited heat accumulation during the cooling stage for printing temperatures lower than 240 °C, and more specifically for 210 °C.

The cooling stage is captured from the IR measurements along the large strip when the nozzle stays on the blocs. At mid-distance from blocs, the thermal cycling is captured through the time evolution of the filament temperature at this location (Figure 4a). The thermal signature of the PETG sample printed at the highest printing temperatures (255 °C) exhibits larger temperature peaks than the one printed at 220 °C. The difference between the peaks reaches in the average 20 °C. There is no marked difference between the two conditions at the stage of raft production. This stage corresponds to the first 150 s of IR recording. The largest peaks recorded of about 160 °C are distant by 15 s each for the highest printing temperature. The analysis of the average peak and ground temperatures for all thermal cycles is shown in Figure 4b for all printing temperature. There is a regular increase of the peak temperature from 145 °C up to 171 °C when the printing temperature is increased in the full range.

The average ground temperature seems constant for all printing conditions mostly because of the large cooling rates taking place during the laying down process. This means that the peak temperature is the main indicator of the heat accumulation within the printed PETG filament. The difference between the ground and peak temperatures varies from 108 °C to 131 °C. Such a difference demonstrates a significant cooling for toolpath distances as large as 60 mm.

For more complex PETG parts such as dog-bone specimens, all printing attempts of these features is inconclusive using PETG for printing temperatures lower than 240 °C. This means that when the difference between the ground and peak temperatures is lower than 120 °C, PETG printability is compromised. These results compare fairly with the printing conditions reported in the literature. For instance, Nieto et al. [19] selected 240°C as a printing temperature for PETG using another type of printing additive manufacturing (AM) called pellet-based AM.

Table 2 compares the densities of the as-received and printed PETG. According to the data sheet, the density of PETG is given as a range of values. Oladapo et al. [21] report a single value of 1.27 g/cm^3^, which lies within the range provided in this study.

The analysis of the overall density of the PETG prints derived from the weight and volume measurements shows that there is up to 19% of density reduction when the dog-bone specimens are printed at 240 °C. The reduction in overall density is limited to 16% when the highest printing temperature is selected. Based on the density of the as-received PETG (the mid-value 1.27 g/cm^3^ is taken as a reference, the porosity content derived, under these terms, varies between 16% and 19%. The nature and extent of this porosity is discussed further more through X-ray micro-tomography imaging.

### 4.2. Mechanical Response of 3D Printed PETG

Figure 5 shows the mechanical results of the as-received PETG filament, which correspond to the tensile experiment shown in Figure 1b. A large extension prior failure is the main characteristic revealed in Figure 5a with a significant variability of the elongation at break (Table 2).

This variability can be explained by the randomly distributed surface flaws that significantly trigger the failure of the filament. A lower elongation at break is reported from the material data sheet (between 102% and 118%). In addition, the tensile response of PETG is characterised by a limited plasticity stage. The derived engineering constants are summarised in Table 2. The measured Young’s modulus and tensile strength are both smaller compared to the values from the datasheet (between 2.01 GPa and 2.11 GPa for Young’s modulus and between 60 and 66 MPa for tensile strength). It has to be mentioned that the variability of the results obtained in this study is fairly acceptable (12% for Young’s modulus and only 2% for tensile strength). Hamidi et al. [20] report 1.74 GPa for Young’s modulus and 46 MPa for tensile strength in the case of PETG reinforced with CNT fibres. In Figure 5b is shown the effect of the FDM processing on the tensile performance of PETG. One significant feature is the substantial reduction of the elongation at break, which is observed irrespective of the printing temperature. In this regard, this is a clear demonstration of the effect of the filament arrangement on the tensile response of PETG. Further explanation is provided from finite element results. Figure 5b also suggests a gradual failure because the difference between the strain at peak stress and the elongation at break is significant (up to 0.05). There is, in addition, no major difference between the printing conditions in terms of slope and a limited improvement of the peak stress is observed when the highest printing temperature is selected. This observation is confirmed for both neat and notched specimens. The extracted engineering constants summarised in Table 2 are used to estimate the mechanical loss due to FDM processing. This loss lies within the ranges (38%–40%), and (37%–41%) for Young’s modulus and tensile strength. It is, however, more significant for the elongation at break. Szykiedans et al. [18] report Young’s moduli between 358 MPa and 1477 MPa for PETG reinforced with Z-glass and explain this huge difference by the mechanical anisotropy subsequent to FDM processing. Hamidi et al. [20] report tensile strength magnitudes between 27 MPa and nearly 35 MPa for PETG reinforced by CNT. In this study, the amount of improvement observed when the printing temperature is increased to 250 °C represents only 5% for tensile strength. A slight loss of 3% for Young’s modulus is depicted, which suggests that the printing temperature slightly affects the tensile strength. If the results related to the fracture toughness are considered, the optimal printing temperature is 250 °C. At this temperature, the fracture toughness of the as-received PETG is nearly restored (a loss of 1% for 250 °C against 14% for 240 °C).

The examination of the deformed PETG during tensile loading (Figure 6a) shows the appearance of a necking at the centre of the specimen prior its failure.

This necking can be correlated to the reduction in the filament section within the raster and to the localised stretching of the external frame. These two phenomena can be correlated to the reduction in section of the as-received filament upon large stretching (Figure 1b). For notched specimens, the crack departure from the notch is responsible for the failure of the 3D printed PETG. The crack, in addition, seems to follow the raster orientation. A large deviation from the opening mode is observed with typical crack propagation angles of –45° or +45° with respect to the loading direction. These are exactly the angles of filament crossing within the plane of construction. It can be concluded that a significant shearing mode is involved in the failure of PETG.

The ranking of PETG with respect to other polymers used in FDM is depicted in Figure 7 as a function of the printing temperature. This comparison is based on data collected by the authors for other polymers that were processed under the same printing conditions. The printing temperature range is adjusted according to the printability of each feedstock material. In terms of stiffness (Figure 7a) and tensile strength (Figure 7b), PETG exhibits an acceptable performance similar to the one of acrylonitrile styrene acrylate (ASA) and PLA-PHA (polylactic acid-polyhydroxyalkanoates). It is, however, lower compared to the performance of PLA-hemp processed at lower printing temperatures. The fracture toughness of 3D printed PETG lies at the middle range of the polymer collection and exhibits a similar ranking to PLA-PHA and PLA-Hemp processed at the lowest printing temperatures (Figure 7c).

### 4.3. Structural Analysis of 3D Printed PETG

Figure 8 shows key features of the PETG ductile fracture observed by a scanning electron microscope (SEM). Figure 8a depicts gaps of about 80 µm between adjacent filaments. The same figure illustrates the significant stretching of PETG filaments that is experienced during tensile loading. Zoomed views (Figure 8b,c) of the fractured filament suggest that this stretching is also combined with shearing because the tearing direction on each individual filament is misaligned with the loading direction. Figure 8d,e show that the filament stretching is localised. Some filaments are not strongly affected by section reduction and their extension in the longitudinal direction is limited (on the right in Figure 8d and on the left in Figure 8e).

Finally, Figure 8f reveals a quasi-brittle fracturing of the external frame. The filaments composing the external frame are aligned along the loading direction. They confer structural stability to the printed PETG sample by connecting the filaments crossed within the plane of construction. These filaments are subject to pure tensile loads and experience limited shearing compared to the rest of the PETG structure.

Exploration of the microstructure generated by FDM processing is provided by X-ray micro-tomography results. Figure 9a shows the main cross-section views in XY (plane of construction), XZ, and YZ planes. The process-generated porosity and the surrounding air are identified by dark grey levels. The signature of the filament arrangement in the plane of construction can be deduced from the pore connectivity in −45°/+45° directions. The typical size of the porosity is 430 µm and can extend over the millimetre scale (1.44 mm in some regions). There is no distinct porosity generated near the external frame. This could be expected from the rapid change of toolpath trajectory near the boundaries. In the planes (XZ, YZ) containing the building direction (Z), no major connectivity between the pores is observed. These are all aligned in the building direction and represent the result of the cumulative gap left between adjacent filaments. The surface finishing state can be approached from the XZ view. For instance, the measurement of the average roughness along the length is 98 µm and the maximum roughness is 328 µm. It is worth mentioning that the roughness generated by the FDM processing is half the layer height (0.2 mm).

In the YZ plane, the roughness along the width direction is similar to the one measured along the length (average roughness is 93 µm and maximum roughness is 306 µm). Figure 9b shows perspective views of the main features of 3D printed PETG. These views highlight, for instance, that the external frame is composed of two adjacent filaments. Within the raster, there is a visible alteration of the filament geometry near the external frame due to the considerations about the abrupt change in the toolpath trajectory. The top view depicts the alternation of the filament arrangements along the thickness in a sequence of +45°/–45°. The top surface also reveals a large surface porosity due to the gap between the filaments on the top layer. This porosity is expected to be larger compared to the bulk porosity. The front view shows that the layers are regular in thickness. Finally the internal porosity exhibits a low connectivity in the two main directions (width and length).

The porosity profiles derived from the 3D image segmentation are shown in Figure 10. The total amount of porosity corresponding to the 3D microstructure in Figure 9 is 2.18%. This amount is associated to PETG printed at 250 °C. The comparison between this porosity content with the one derived from density measurement (Table 2) highlights a difference of about 5%. This large difference can be explained by the fact that the porosity measured from 3D imaging is a bulk porosity that does not include the amount of surface porosity. In addition, porosities of a size lower than the voxel size of 14 µm cannot be captured using X-ray micro-tomography.

All porosity profiles in Figure 10a show regular alternation between low and high porosity contents. This alternation is related to the regularity of the porous network. Along the width, there is, in addition, a larger perturbation in the porosity profile, which goes from the ground values at the external frame to peak values as large as 4% within the volume.

The perspective view of the porous network (Figure 10b) confirms the large connectivity of the porosity in the building direction and the limited one along the width and length of the printed PETG sample. Due to the lack of pore percolation in the X and Y directions, the largest connecting pore has a volume of 0.15 mm^3^, and represents only 1% of the total porosity volume.

### 4.4. Prediction of 3D Printed PETG Performance

Finite element computation based on the implementation of 3D images of the 3D printed PETG microstructure are discussed. Finite element results related to the prediction of the elasticity behaviour of 3D printed PETG are shown in Figure 11. Computations performed using different resolutions are compared. In Figure 11a, the resolution is expressed as a voxel size, where a large voxel size corresponds to a small regular meshing and vice versa. It has to be mentioned that the original resolution for which the voxel size is only 14.31 µm cannot be reached with the present computation resources. The nodal results expressed as a stress intensity counterplots are retrieved for each voxel size. The stress intensity is defined as:
(2)$$\sigma_{I}\left(MPa\right)=MAX\left(\left|\sigma_{1}-\sigma_{2}\right|,\left|\sigma_{1}-\sigma_{3}\right|,\left|\sigma_{2}-\sigma_{3}\right|\right)$$
where $\sigma_{1}$, $\sigma_{2}$, $\sigma_{3}$ are the principal stress values.

Figure 11a compares the stress intensity distributions for different voxel sizes ranging from 24 µm up to 143 µm. The same stress scale between 2 and 13 MPa is applied to all counterplots to ease the reading of the results. The FE predictions, for the lowest resolution (voxel size of 143 µm), demonstrate the presence of a heterogeneous stress field that differentiates between two main regions, namely the raster and the external frame.

The stress levels are larger and more homogeneous at the external frame compared to the levels within the raster. This indicates that the external frame plays an important role to improve the load bearing capabilities of the printed PETG.

The low resolution does not capture the effect of the layered structure. For larger resolutions, the effect of the printed layers on the stress field is observed more particularly for voxel sizes larger than 72 µm. The alternation of low and high stress levels within the raster confirms the inhomogeneous stretching observed through SEM micrographs in Figure 8. Because the filaments are misaligned with respect to the loading direction, the loading tends to force them to align. This has two main consequences. The first consequence is the development of local shearing especially at the joints where high stress levels are predicted (Figure 11a). The second consequence is that the longitudinal extension of the filaments is only possible between the joints. Figure 11b shows the predicted Young’s modulus as a function of the voxel size. There is no major difference between the computed property as most of the variation is captured between 507 MPa and 524 MPa. The stability of Young’s modulus is explained by the limited change in the proportion of the solid phase when the resolution is varied. The meaningful difference between the tried voxel sizes is related to the stress localisation that is only captured when large resolutions are implemented in the FE model. If the predicted Young’s modulus at the largest voxel size (507 MPa) is compared with the experimental value (478 MPa) for the printing temperature 250 °C, the model overestimates by 6% the stiffness of the printed PETG. This overestimation can be related to an overestimation of the solid phase proportion or to a limited load transfer between the filaments. In this last scenario, contact models or interfacial properties can be added to adjust the predicted modulus. Due to the limited difference between the predicted and experimental values, it can be stated that the model fairly capture the main deformation mechanisms in the 3D printed PETG.

### 4.5. Printability of Technical Parts Using PETG

The former results indicate that 250 °C can be considered as an optimal printing temperature for PETG. With regard to the bed temperature, this parameter has a significant effect on the adhesion between the platform and the first layers of the print. Since our objective is to lower as much as possible the electric energy used to print PETG, this parameter is kept constant in the study (Table 1). To overcome the difficulty to print technical parts with a low bed temperature, the use of a raft helps to make the junction between the first layers of the print and the bed. This allows maintaining sufficient cohesion between the first layers and avoid premature decohesion. Figure 12a shows the setup used to monitor the printing process for two technical parts exhibiting different dimensions. The thermal behaviour is expected to vary depending on the length of the toolpath trajectory (Figure 12b). Figure 12b shows the final rendering of the technical parts with the raft. The technical parts remain sticky to the platform and bending at the corners is significantly reduced even for thicker parts (low warping).

The infra-red camera measurements are detailed in Figure 12c,d. The temperature profiles for each part are captured at the final stage of processing where it is possible to measure the temperature distribution along the depth and length. The analysis shows that the temperature profile at the bottom layer of the smallest part (Figure 12c) maintain a temperature as large as 53 °C at the centre and the temperature gradually decreases along the length of 60 mm down to 40 °C at the edges. A temperature drop of at least 10 °C is observed for the largest part (Figure 12d) along the same bottom line. At the top layer, the peak temperature corresponds to the filament temperature at the nozzle tip position. For the sake of comparison, the temperature profiles for both parts are captured when the nozzle tip is close to the centre position. The asymmetric temperatures profiles provide some clues about the direction of nozzle motion, which is for both parts, from left to right. On the left side of the peak temperature, a difference of about 20°C between the parts is caused by a ratio of part lengths of 60/100. In addition, the in-depth temperature profiles show a temperature gradient that varies significantly depending on the part and the longitudinal position. At the edge, the cooling rate is approximately 1 °C/mm and 2 °C/mm for the smallest and largest parts, respectively. It increases to 15 °C/mm and 26 °C/mm at the centre position for the same parts.

The former technical parts are printed using one of the available PETG brand (3DFilTech) in the market. Other brands such as RS EZ-GLAZE from RS Pro or the PETG filament from SUNLU LTD would fit low printing temperatures (according to the suppliers) within the range (195–225 °C) and with a bed temperature of 55 °C. Other types of PETG filaments such as the ones provided by ColorFabb company require higher printing temperatures (235–255 °C) and a higher bed temperature (between 70 °C and 80 °C). The versality of the PETG printability conditions and the wide range of characteristics makes it difficult to achieve the same rendering and the same performance of the technical parts shown in Figure 12 without studying the thermal conditions during the laying down process and optimising these with regards to the target application.

## 5. Conclusions

This study concludes that despite a low glass temperature of 77 °C, the FDM processing of PETG requires high printing temperature above 230 °C. Printability under lower temperatures compromises the adhesion between the printing platform and the PETG filament. The infrared measurements reveal that the heat accumulation during thermal cycling of PETG is only captured through the peak temperature, which increases up to 160 °C for a printing temperature of 255 °C. Dog-bone specimens printed within the printing temperature range (240–250 °C) exhibit a low overall density due to the presence of process-induced porosity. The amount of porosity varies between 16% and 19% based on the density measurement. X-ray micro-tomography results contrast with the density measurements since the 3D image processing suggests a bulk porosity content of only 2% for a printing temperature of 250 °C. This difference can be explained by the contribution of surface porosity, which accounts for a rough finishing state and the presence of an average roughness of about 100 µm. In addition, a low connectivity is measured between the pores within the plane of construction. The pore connectivity only extends thorough the thickness of the printed PETG.

From the mechanical viewpoint, FDM processing of PETG results in a substantial reduction of the elongation at break and an evident loss in both stiffness (up to 40%) and tensile strength (up to 40%). The property that is restored by FDM processing is the fracture toughness. Despite the relative improvement of the tensile performance when PETG is printed at 250 °C, the loss in mechanical performance still represents one-third of the raw material for both stiffness and tensile strength. This study demonstrates that the fracture properties are correlated with the filament arrangement within the plane of construction as indicated by the deviation of the crack propagation. Finite element predictions reveal the development of a heterogeneous stress field within the raster and more regular stress levels are observed at the external frame. In this regard, the FE predictions demonstrate the lead role of the external frame as a load bearing element that exhibit the highest stress. Deformation mechanisms pointed out through the FE simulations combine local shearing and inhomogeneous stretching that appear correlated to the presence of joints between the filaments and the misalignment of the filament with respect to the loading direction.

## Figures and Tables

**Figure 1 polymers-11-01220-f001:**
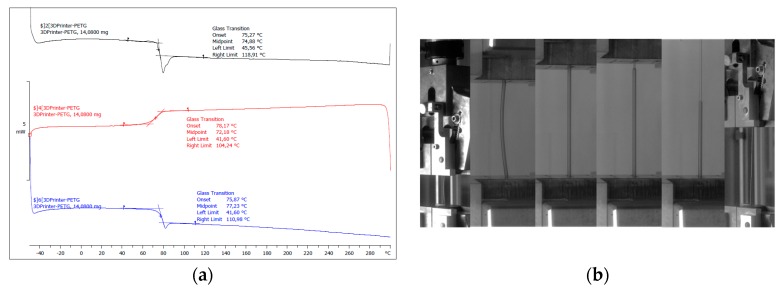
(**a**) Thermal analysis of the as-received polyethylene terephthalate glycol (PETG) filament using differential scanning calorimetry, (**b**) Tensile testing performed on a PETG single filament.

**Figure 2 polymers-11-01220-f002:**
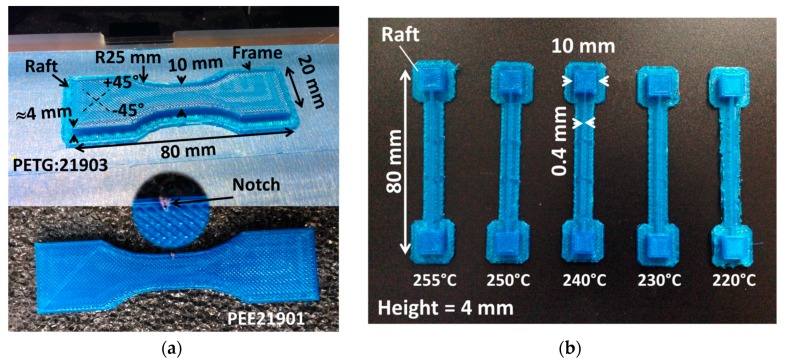
Some of the printed specimens in PETG considered in this study (**a**) dog-bone specimens for tensile testing, (**b**) specimens manufactured during the thermal analysis of the laying down process for an increasing printing temperature.

**Figure 3 polymers-11-01220-f003:**
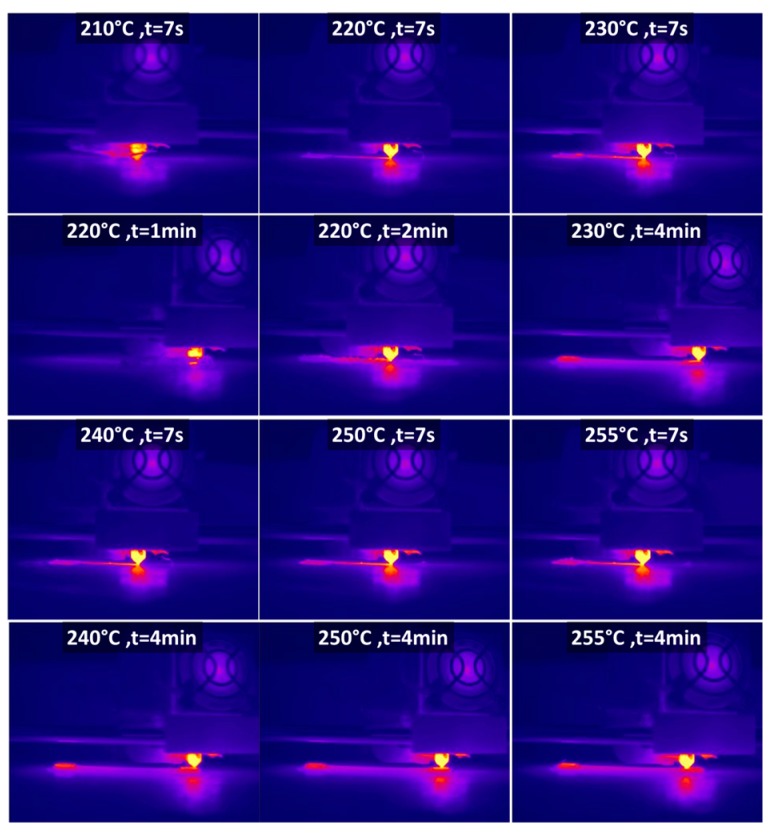
Comparison between the PETG filament temperature during the fused deposition modelling (FDM) process for a wide range of printing temperatures.

**Figure 4 polymers-11-01220-f004:**
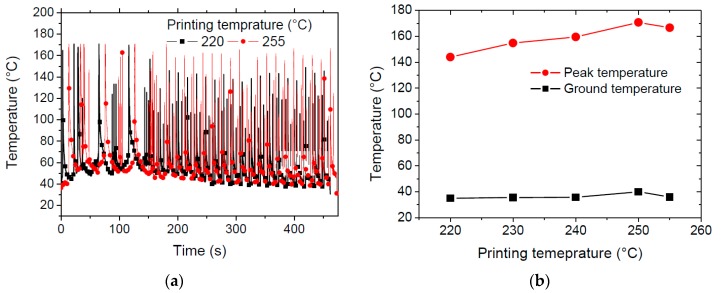
Thermal cycling captured during the laying down of PETG filament using infra-red camera. (**a**) evolution of filament temperature at the centre of the specimen for the minimum and maximum printing temperatures, (**b**) The peak and ground temperatures for all considered FDM conditions.

**Figure 5 polymers-11-01220-f005:**
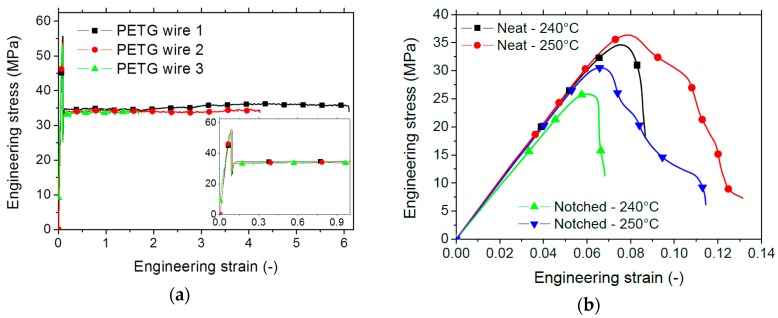
Tensile response of the (**a**) as-received and (**b**) printed PETG as a function of the printing temperature.

**Figure 6 polymers-11-01220-f006:**
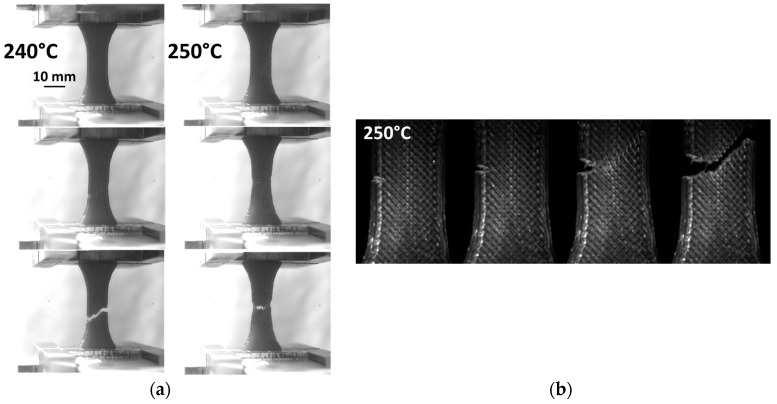
(**a**) Deformation sequences showing localisation in neat PETG specimens, and (**b**) crack propagation trajectory in notched specimen printed at 250 °C.

**Figure 7 polymers-11-01220-f007:**
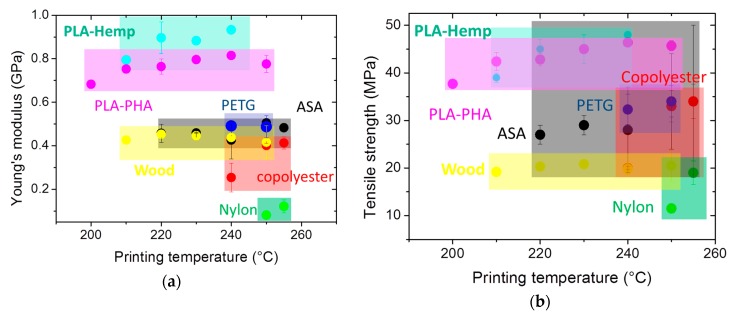
Comparison between tensile performance of less known polymers in FDM with PETG considered in this study as a function of the printing temperature. (**a**) Young’s modulus, (**b**) tensile strength, and (**c**) fracture toughness.

**Figure 8 polymers-11-01220-f008:**
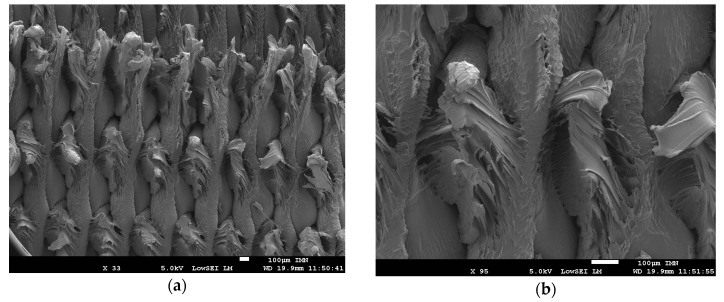
Analysis of the microstructural arrangement near the fractured zones in printed PETG using a scanning electron microscope. The printing temperature is 240 °C. (**a**) gaps between adjacent filaments, (**b**) stretched filaments, (**c**) zoomed view on inhomogeneous filament stretching, (**d**) localised deformation, (**e**) zoom-in on localisation in deformed raster, (**f**) brittle fractured external frame.

**Figure 9 polymers-11-01220-f009:**
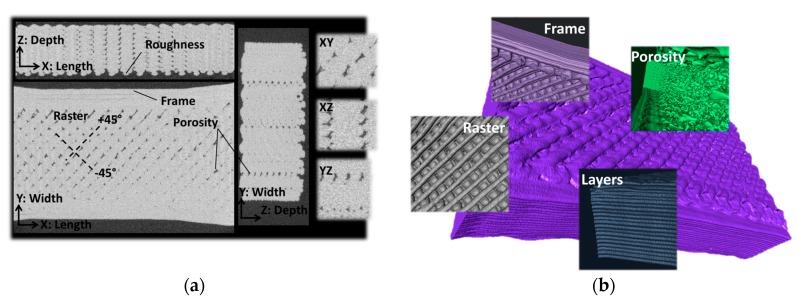
(**a**) Cross-section views showing the microstructural arrangement in the 3D printed PETG investigated using X-ray micro-tomography, (**b**) Illustrative views of the main features revealed from image processing including the layered structure, the process-induced porosity, the filament arrangement within the plane of construction and the external frame.

**Figure 10 polymers-11-01220-f010:**
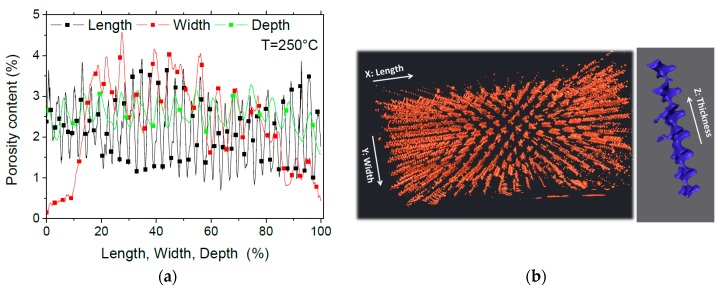
(**a**) Porosity profiles in the main directions and (**b**) perspective view of the porous network in 3D printed PETG.

**Figure 11 polymers-11-01220-f011:**
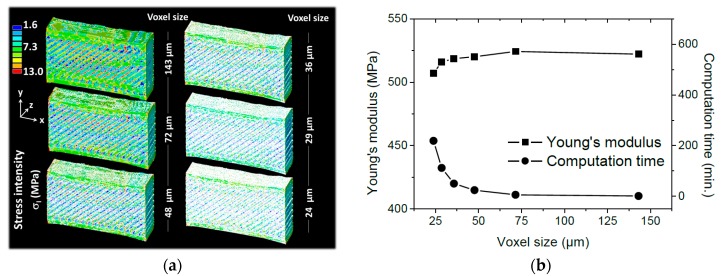
Illustration of finite element results (**a**) predicted stress intensity counterplot and (**b**) effective Young’s modulus of 3D printed PETG as a function of the voxel size.

**Figure 12 polymers-11-01220-f012:**
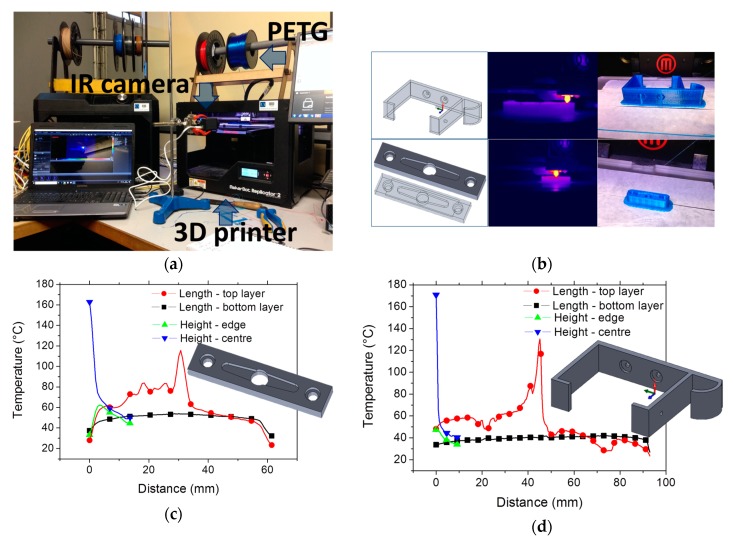
(**a**) Setup to monitor technical part printing using PETG, (**b**) CAD models, Ongoing processed captured using IR camera and final printed parts, thermal cycles at particular positions for (**c**) small and (**d**) large technical parts.

**Table 1 polymers-11-01220-t001:** Process conditions for 3D printing of PETG.

Parameter	Value
support density	0.2
support to model spacing	0.4 mm
support angle	68°
support layer thickness	0.2 mm
printing temperature	210 °C to 255 °C
bed temperature	25 °C
printing speed	150 mm/s
Z-axis speed	23 mm/s
sample orientation	Building // thickness
layer height	0.2 mm
filament diameter	1.75 mm
infill density	100%
nozzle diameter	0.4 mm
layup	+45°/−45°
raft to model spacing	0.35 mm
base pattern spacing in raft	0.8 mm
base pattern length in raft	15 mm
base layer density in raft	0.7
base layer height in raft	0.3 mm
raft margin in raft	4 mm

**Table 2 polymers-11-01220-t002:** Comparison between the physical properties of as-received and printed PETG.

Material	Printing Temperature °C	Density (g/cm^3^)	Porosity content * (%)	Young’s Modulus (MPa)	Tensile Strength (MPa)	Elongation at Break (%)	Fracture Toughness MPa × m1/2
as-received	-	1.26–1.28	0	791 ± 91	54.3 ± 1.3	400 ± 220	2.11–2.54
printed	240	1.03 ± 0.01	19 ± 0.9	491 ± 13	32.3 ± 3.2	8.8 ± 0.2	1.81 ± 0.01
	250	1.07 ± 0.02	16 ± 1.6	478 ± 51	34.0 ± 3.3	11.7 ± 2.0	2.09 ± 0.01

* based on density measurements.

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
