# Peer review of "Printability and Tensile Performance of 3D Printed Polyethylene Terephthalate Glycol Using Fused Deposition Modelling"

_polymers, 2019, doi:10.3390/polym11071220_

Reviewer 1 Report

The authors performed characterizing the FDM 3D printed PETG filament structure using several methods. PETG is a stronger and durable material than ABS, yet it is simple to print as PLA.  It is well known that PETG can be printed from 220-250 degree Celsius. However, studies addressed 3D printed structure using PETG material is limited. Authors addressed print temperature and its impact on the tensile performance using 3Dfiltech. It is good work for the 3d printing field. The paper should be considered for publication after minor corrections.

1.      Based on my experience, the print temperature varies between PETG brands. Authors build the entire paper based on just one filament observations. Authors should provide comments on the impact of results for other PETG brands.

2.      Fig 4a and Fig 4b are identical. Check it carefully

3.      Fig 7b and 7c are identical, yet different captions were provided.

4.      Fig 8e and Fig 8f are identical.

Author Response

Reviewer #1

The authors performed characterizing the FDM 3D printed PETG filament structure using several methods. PETG is a stronger and durable material than ABS, yet it is simple to print as PLA.  It is well known that PETG can be printed from 220-250 degree Celsius. However, studies addressed 3D printed structure using PETG material is limited. Authors addressed print temperature and its impact on the tensile performance using 3Dfiltech. It is good work for the 3d printing field. The paper should be considered for publication after minor corrections.

 We thank the reviewer for his positive opinion about our paper. He will find hereafter our detailed answer to his comments.

1.      Based on my experience, the print temperature varies between PETG brands. Authors build the entire paper based on just one filament observations. Authors should provide comments on the impact of results for other PETG brands.

The reviewer is right. We have added the following discussion about the use of other brands.  In page 14: ‘’ The former technical parts are … with regards to the target application. “

2.      Fig 4a and Fig 4b are identical. Check it carefully

We are really sorry for the confusion. A copy-paste from the figure file went wrong, in the new version the right Fig. 4b is included.

3.      Fig 7b and 7c are identical, yet different captions were provided.

Here again the same issue, we thank the reviewer or this comment. The right figure Fig. 7c is now inserted.

4.      Fig 8e and Fig 8f are identical.

 Fig. 8f is now inserted

Reviewer 2 Report

Thank you for submitting this paper. Here are some observations:

- the introduction about FDM on the first page is not really needed and not entirely tru as there are other constructive types suh as FDM printers that move the platform on the vertical axis instead of the nozzle system

- Table 1 presents the bed temperature at 25 degrees Celsius, but the filament supplier recommends temperatures between 40 and 65 degrees Celsius - please explain this difference and the motivation for this

- please explain whether the specimens used for testing are created according the the standard ASTM D638

- the specimens were printed only at 240 and 250 degrees Celsius and not as claimed in the beginning of the paper "in a wide range of printing temperatures from 210°C to 255°C"

- the authors should consider subsections for section 3 Results, in order to improve the  paper readability

- the FEA is not clear enough (page 12); please explain it further with more details

- the last subsection of the 3. Results section (on page13)  is not clearly explained - does the bed temperature influence the results (i think it does, but the authors are not explaining this issue)

 - Conclusions sections should be number 4 instead of 5

Author Response

Comments and Suggestions for Authors

Thank you for submitting this paper. Here are some observations:

- the introduction about FDM on the first page is not really needed and not entirely true as there are other constructive types suh as FDM printers that move the platform on the vertical axis instead of the nozzle system

We withdrawn this section as recommended by the reviewer. We also agree that there are several FDM systems. The one described in the former version is one of these versions.

Amendment in page 1: “In Fused Deposition Modelling (FDM),…trajectory [4].” 

- Table 1 presents the bed temperature at 25 degrees Celsius, but the filament supplier recommends temperatures between 40 and 65 degrees Celsius - please explain this difference and the motivation for this

Our motivation is to produce 3D printed parts using the lowest energy consumption possible. This motivation led to the selection of smaller printing temperatures in the range (220-250°C) and a low platform temperature. We believe that this challenge was addressed successfully as we were able to print technical part under low energy constrains.

Justification is now added in page 2: “Among the selected conditions,… lowest energy consumption.”

- please explain whether the specimens used for testing are created according the standard ASTM D638

The CAD models for the dog-bone specimens were created using the norm ISO 527-1/2. See for instance the following reference

ISO, S., 527-1. Plastics. Determination of tensile properties. Part, 2012. 1.

Amendment in page 2.

- the specimens were printed only at 240 and 250 degrees Celsius and not as claimed in the beginning of the paper "in a wide range of printing temperatures from 210°C to 255°C"

This was the aim to check the printability of the specimens within the large range. See for instance the printed specimens in Fig. 2b. however, the attempts to print tensile specimens at temperatures lower than 240°C failed.  To comment further on this aspect, we suggested the following sentence in page  6: “For more complex PETG parts such as dog-bone specimens, all printing attempts of these features is inconclusive using PETG for printing temperatures lower than 240°C. This means that when the difference between the ground and peak temperatures is lower than 120°C, PETG printability is compromised.”

- the authors should consider subsections for section 3 Results, in order to improve the paper readability

We followed the recommendation of the reviewer. Now sub-sections are added to add more clarity to the manuscript.

- the FEA is not clear enough (page 12); please explain it further with more details

We introduced more clarification about the FEA in the new version. If there is any specific point that the reviewer would like to highlight we would be happy to address it.

- the last subsection of the 3. Results section (on page13)  is not clearly explained - does the bed temperature influence the results (i think it does, but the authors are not explaining this issue)

The reviewer is right, the bed temperature has a significant effect on the adhesion between the platform and the first layers of the print. Since our objective was to lower as much as possible the electric energy used to print PETG, we kept this parameter constant in the study. To overcome the difficulty to print with a low bed temperature, we added the raft that make the junction between the first layers of the print and the bed. This allowed us to maintain sufficient cohesion between the first layers and avoid premature decohesion.

Amendment in page 13: “With regards to the bed temperature,… premature decohesion” + “Figure 12b shows the… for thicker parts.”

 - Conclusions sections should be number 4 instead of 5

This is corrected now and the numbering of all sections was checked and corrected accordingly.
